# Transcriptome Analysis Revealed the Potential Molecular Mechanism of Anthocyanidins’ Improved Salt Tolerance in Maize Seedlings

**DOI:** 10.3390/plants12152793

**Published:** 2023-07-27

**Authors:** Jie Wang, Zhipeng Yuan, Delin Li, Minghao Cai, Zhi Liang, Quanquan Chen, Xuemei Du, Jianhua Wang, Riliang Gu, Li Li

**Affiliations:** 1Beijing Innovation Center for Crop Seed Technology, Ministry of Agriculture and Rural Affairs, College of Agronomy and Biotechnology, China Agricultural University, Beijing 100193, China; wangjie_stub@163.com (J.W.); iuenzipo@163.com (Z.Y.); delin.bio@gmail.com (D.L.); s20193010101@cau.edu.cn (M.C.); liangzhinzz@163.com (Z.L.); znchenquanquan@163.com (Q.C.); duxuemei1986@yeah.net (X.D.); wangjh63@cau.edu.cn (J.W.); 2Tropical Crops Genetic Resources Institute, Chinese Academy of Tropical Agricultural Science, Haikou 571101, China; 3Sanya Research Institute, Chinese Academy of Tropical Agricultural Science, Sanya 572000, China; 4Sanya Institute, China Agricultural University, Sanya 572025, China; 5Institute of Crop Sciences, Chinese Academy of Agricultural Sciences, Beijing 100081, China

**Keywords:** maize, seedling growth, salt stress tolerance, anthocyanidin, transcriptome sequencing

## Abstract

Anthocyanin, a kind of flavonoid, plays a crucial role in plant resistance to abiotic stress. Salt stress is a kind of abiotic stress that can damage the growth and development of plant seedlings. However, limited research has been conducted on the involvement of maize seedlings in salt stress resistance via anthocyanin accumulation, and its potential molecular mechanism is still unclear. Therefore, it is of great significance for the normal growth and development of maize seedlings to explore the potential molecular mechanism of anthocyanin improving salt tolerance of seedlings via transcriptome analysis. In this study, we identified two W22 inbred lines (tolerant line pur–W22 and sensitive line bro–W22) exhibiting differential tolerance to salt stress during seedling growth and development but showing no significant differences in seedling characteristics under non–treatment conditions. In order to identify the specific genes involved in seedlings’ salt stress response, we generated two recombinant inbred lines (RIL^pur–W22^ and RIL^bro–W22^) by crossing pur–W22 and bro–W22, and then performed transcriptome analysis on seedlings grown under both non–treatment and salt treatment conditions. A total of 6100 and 5710 differentially expressed genes (DEGs) were identified in RIL^pur–W22^ and RIL^bro–W22^ seedlings, respectively, under salt–stressed conditions when compared to the non–treated groups. Among these DEGs, 3160 were identified as being present in both RIL^pur–W22^ and RIL^bro–W22^, and these served as commonly stressed EDGs that were mainly enriched in the redox process, the monomer metabolic process, catalytic activity, the plasma membrane, and metabolic process regulation. Furthermore, we detected 1728 specific DEGs in the salt–tolerant RIL^pur–W22^ line that were not detected in the salt–sensitive RIL^bro–W22^ line, of which 887 were upregulated and 841 were downregulated. These DEGs are primarily associated with redox processes, biological regulation, and the plasma membrane. Notably, the anthocyanin synthesis related genes in RIL^pur–W22^ were strongly induced under salt treatment conditions, which was consistented with the salt tolerance phenotype of its seedlings. In summary, the results of the transcriptome analysis not only expanded our understanding of the complex molecular mechanism of anthocyanin in improving the salt tolerance of maize seedlings, but also, the DEGs specifically expressed in the salt–tolerant line (RIL^pur–W22^) provided candidate genes for further genetic analysis.

## 1. Introduction

Maize is the largest grain product in the world, renowned for its rich protein, starch, vitamin, and trace element content [1]. Maize exhibits various colors, including yellow, white, purple, and red [2]. The different colored maize seeds contain different kinds of pigments [3]. For example, the pigments in yellow maize are mainly carotene and riboflavin, and the pigments in purple and red maize are mainly anthocyanins [4,5,6]. As awareness grows regarding the nutritional and functional health benefits of anthocyanins, there has been an increased demand for maize varieties enriched with these pigments [7,8].

Anthocyanin is a natural water–soluble pigment and one of the flavonoid compounds widely found in nature [9]. Enzymes such as phenylalanine ammonialyase (PAL), cinnamate 4–hydroxylase (C4H), 4–coumarate CoA ligase (4CL), chalcone synthase (CHS), chalcone isomerase (CHI), flavanone 3–hydroxylase (F3H), flavonoid 3′–hydroxylase (F3′H), flavonoid 3′5′–hydroxylase (F3′5′H), dihydroflavonol 4–reductase (DFR), anthocyanin synthase (ANS), and flavonoid 3–*O*–glucosyltransferase (UFGT) are involved in the anthocyanin biosynthesis pathway [10,11,12,13,14,15,16]. Meanwhile, v–myb avian myeloblastosis viral oncogene homolog (MYB), basic helix–loop–helix (bHLH), WD–repeat protein (WD40), and other transcription factors can affect the synthesis of anthocyanins by regulating the expression of structural genes [17,18,19]. Furthermore, environmental factors also impact the accumulation of anthocyanins in plants [20,21,22,23,24].

The global area of saline–alkali land is about 950 million hectares. Saline soil will directly affect seed germination, seedling morphogenesis, and plant growth and development, thus affecting crop yield [25,26,27]. Under environmental stress, plants can regulate the expression of structural genes and regulatory genes in the anthocyanin synthesis pathway to accumulate anthocyanins in plants and thus withstand the damage caused by environmental stress [28,29,30,31]. Especially under salt stress, in one study, *AtDFR* expression in *Brassica napus* L. was upregulated, anthocyanidin accumulation was increased, and plant tolerance to salt stress was enhanced [31]. Therefore, increasing the anthocyanin content of maize seedlings and enhancing their salt stress tolerance are essential for maize production.

A transcriptome refers to the sum of all gene transcription products of a specific organism at specific development stage. Transcriptome analysis is a powerful tool for identifying genes that contribute to complex traits [32,33,34,35]. The molecular regulation mechanism of flavonoid biosynthesis under salt stress has been reported in plants such as sorghum [36], radish [37], grape [38], and alfalfa [39]. Many differentially expressed genes (DEGs) related to seedling growth via their role in regulating flavonoid biosynthesis under salt stress have also been identified. However, there is limited research on the role of anthocyanins in conferring salt stress resistance in maize seedlings. In this study, we used seedlings of two W22 inbred lines (RIL^pur–W22^ and RIL^bro–W22^) for transcriptome analysis. Expected to explore the differentially expressed genes (DEGs) involved in the salt tolerance of maize seedlings via transcriptome analysis, so as to further understand the potential molecular mechanism of anthocyanin in improving the salt tolerance of maize seedlings.

## 2. Results

### 2.1. The Purple–Colored W22 and Bronze–Colored W22 Show Different Seedling Growth under Salt Stress

The seeds of purple W22 (pur–W22) inbred lines were purple, while the seeds of bronze W22 (bro–W22) inbred lines were bronze. Upon comparing the seed traits of the two W22 inbred lines, it was observed that, apart from the color difference, the seed length, width, and thickness of the pur–W22 inbred lines were significantly larger than those of bro–W22 (Appendix A). Additionally, bro–W22 displayed a significant decrease in 100–grain weight (Appendix A). To evaluate the salt stress tolerance of the pur–W22 and bro–W22 lines, we subjected them to a 100 mM NaCl solution to simulate salt stress. Under salt–treated conditions, bro–W22 exhibited shorter seedling length and root length compared to pur–W22. However, no significant differences in seedling length and root length were observed between the seedlings under non–treated conditions. Furthermore, both pur–W22 and bro–W22 seedlings displayed inhibited root and seedling lengths under salt treatment compared to non–treated conditions (Appendix A). In addition, both lines exhibited similar germination rates, exceeding 96%, under both non–treated and salt–treated conditions (Appendix A). By simulating a salt stress environment using sand culture, we corroborated the findings obtained from the germination experiment, indicating that pur–W22 exhibits stronger salt stress tolerance than bro–W22 (Appendix A).

After conducting a cross between pur–W22 and bro–W22, we obtained the F_2_ population through self–pollination. Subsequently, the F_6_ generation RILs (RIL^pur–W22^ and RIL^bro–W22^) were derived by self–pollinating the F_2_ generation plants over multiple generations. Comparing the seed traits of the two RIL populations, we observed no significant differences in seed length, width, thickness, or 100–grain weight between RIL^pur–W22^ and RIL^bro–W22^, except for the variations in color (Figure 1a–e). Furthermore, we observed that RIL^pur–W22^ seedlings exhibited significantly longer seedling and root lengths compared to RIL^bro–W22^ under salt stress conditions. And the seedling length and root length of RIL^pur–W22^ and RIL^bro–W22^ seedlings were inhibited under salt–treated compared with non–treated conditions (Figure 1f–h). It was noteworthy that RIL^pur–W22^ seedlings still showed similar salt tolerance to its parent pur–W22 after the differences in seed length, seed width, seed thickness, and 100–seed weight were removed (Figure 1f–h).

### 2.2. Transcriptome Analysis of RILs undergoing Salt Treatment

To further investigate the impact of salt treatment on the development of seedlings of different seed colors, we selected RIL^pur–W22^ and RIL^bro–W22^ seedlings (consisting of aboveground stems and leaves, with the seeds removed) that had germinated and grown for 14 days under non–treated and salt stress environments (Figure 2a). We performed two biological replicates, totaling eight samples, and extracted total RNA for sequencing. Principal component analysis using the expression data from these samples demonstrated better repeatability between the two biological replicates (Figure 2b). Moreover, we observed significant differences between the transcriptomes of non–treated RIL^pur–W22^ seedlings (NT–RIL^pur–W22^) and salt–treated RIL^pur–W22^ seedlings (salt–RIL^pur–W22^). Similar distinctions were observed between non–treated RILbro–W22 seedlings (NT–RIL^bro–W22^) and salt–treated RIL^bro–W22^ seedlings (salt–RIL^bro–W22^) (Figure 2b). These findings indicate that salt stress had a considerable impact on the growth and development of the seedlings of RIL^pur–W22^ and RIL^bro–W22^.

A total of 369,817,976 clean reads were obtained from transcriptome sequencing. The number of reads per sample ranged from 44,597,506 to 47,933,524, accounting for 95.15% to 97.33% of the total unfiltered reads. Following the alignment of the clean reads to the reference genome of the maize V4 version (Zm–B73–REFERENCE–GRAMENE–4.0), the number of reads and fragment length were calculated for 46,424 genes, which were then assigned FPKM (fragments per kilobase of transcript per million mapped reads) values. Genes with an FPKM value ≥ 1 were considered to be expressed genes. We observed a similar number of expressed genes in non–treated RIL^pur–W22^ and RIL^bro–W22^, while the number of expressed genes in non–treated RILs was lower than that in salt–treated RILs, indicating that there was an induction of gene expression in the salt–stressed environment compared to the non–treated RILs. Notably, the highest number of expressed genes was observed in salt–treated RIL^pur–W22^ seedlings (Figure 2c, Table 1).

### 2.3. Differentially Expressed Genes in RIL^pur–W22^ and RIL^bro–W22^ Seedlings

Using the criteria of fold change ≥2 and a corrected *p*–value of ≤0.01, we screened the DEGs and compared them in various combinations. In the NT–RIL^pur–W22^ vs. NT–RIL^bro–W22^ group, which compares non–treated RIL^pur–W22^ and RIL^bro–W22^ seedlings based on their color difference, we identified 1606 DEGs, including 846 up– and 760 downregulated DEGs. The NT–RIL^pur–W22^ vs. salt–RIL^pur–W22^ group represents DEGs produced by RIL^pur–W22^ seedlings under salt stress, which accounted for 6100 DEGs, with 3109 up– and 2991 downregulated DEGs. Similarly, in the NT–RIL^bro–W22^ vs. salt–RIL^bro–W22^ group, which compares RIL^bro–W22^ seedlings under salt stress, we identified 5710 DEGs, with 2961 up– and 2749 downregulated DEGs. Lastly, the salt–RIL^pur–W22^ vs. salt–RIL^bro–W22^ group represents DEGs resulting from the combined effects of color difference and salt stress between RIL^pur–W22^ and RIL^bro–W22^ seedlings. We found 1040 DEGs, including 578 up– and 462 downregulated DEGs (Figure 2d). Furthermore, the volcano plot (Appendix A) provides a clear and intuitive visualization of the DEGs.

To investigate the genes and pathways influencing the growth and development of RIL^pur–W22^ and RIL^bro–W22^ seedlings in different groups, we conducted Venn diagram analysis on the DEGs from the four groups (Figure 2e). The analysis revealed that 371, 1728, 1475, and 140 DEGs had specific roles in the NT–RIL^pur–W22^ vs. NT–RIL^bro–W22^, NT–RIL^pur–W22^ vs. salt–RIL^pur–W22^, NT–RIL^bro–W22^ vs. salt–RIL^bro–W22^, and salt–RIL^pur–W22^ vs. salt–RIL^bro–W22^ groups, respectively. Moreover, we identified 3160 DEGs that were exclusively co–expressed in the NT–RIL^bro–W22^ vs. salt–RIL^bro–W22^ and NT–RIL^pur–W22^ vs. salt–RIL^pur–W22^ groups. These DEGs were considered to be common salt stress response genes (Figure 2e).

### 2.4. Common Salt–Induced DEGs in RIL^pur–W22^ and RIL^bro–W22^

Out of the 3160 common salt response DEGs identified in RIL^pur–W22^ and RIL^bro–W22^ seedlings, 1637 DEGs were upregulated, while 1521 DEGs were downregulated (Figure 2e, Appendix A). To gain further insights into the functional pathways associated with these common DEGs, we conducted a GO functional enrichment analysis (Appendix A). Among the 1637 upregulated DEGs, significant enrichment was observed in biological processes (BP) related to redox processes (GO: 0055114), monomer metabolic processes (GO: 0044710), transmembrane transport (GO: 0055085), and small molecule metabolic processes (GO: 0044281). The significantly enriched molecular functions (MF) were primarily associated with oxidoreductase activity (GO: 0016491), catalytic activity (GO: 0003824), ion binding (GO: 0043167), and transport activity (GO: 0005215). The significantly enriched cellular components (CC) were linked to the cell periphery (GO: 0071944), plasma membrane (GO: 0005886), and apoplast (GO: 0048046) (Figure 3a, Appendix A). Regarding the 1521 downregulated DEGs, significant categories were mainly associated with biological regulation (GO: 0065007), metabolic process regulation (GO: 0019222), DNA binding (GO: 0003677), and ubiquitin protein transferase activity (GO: 00004842) (Figure 3b, Appendix A). These findings indicate that processes such as oxidoreductase activity, ion binding, catalytic activity, and monomer metabolism play crucial roles in promoting salt stress tolerance in RIL^pur–W22^ and RIL^bro–W22^ seedlings.

To validate the transcriptome results, we selected several DEGs involved in the oxidoreductase activity term and performed qRT–PCR to measure their expression levels in NT–RIL^pur–W22^, NT–RIL^bro–W22^, salt–RIL^pur–W22^, and salt–RIL^bro–W22^ seedlings. Remarkably, the expression patterns of these DEGs were consistent with the transcriptome data, reinforcing the reliability of the transcriptome sequencing results (Figure 3c).

### 2.5. Specific Salt–Induced DEGs in RIL^pur–W22^ and RIL^bro–W22^

In comparison to RIL^bro–W22^ seedlings, RIL^pur–W22^ seedlings exhibited stronger salt tolerance. We identified 1728 DEGs (887 up– and 841 downregulated) that were exclusively expressed in non–treated RIL^pur–W22^ seedlings and salt–treated RIL^pur–W22^ seedlings (NT–RIL^pur–W22^ vs. salt–RIL^pur–W22^) and were not observed in other groups (Figure 2e, Appendix A). Therefore, these genes are considered to be DEGs specifically responding to salt stress in RIL^pur–W22^ seedlings. Additionally, we discovered 1475 DEGs (818 up– and 657 downregulated) exclusively expressed in non–treated RIL^bro–W22^ seedlings and salt–treated RIL^bro–W22^ seedlings (NT–RIL^bro–W22^ vs. salt–RIL^bro–W22^).

We examined the expression patterns of these DEGs in RIL^pur–W22^ and RIL^bro–W22^ seedlings and observed that the specific DEGs in the NT–RIL^pur–W22^ vs. salt–RIL^pur–W22^ group exhibited significant expression changes in RIL^pur–W22^ seedlings under salt stress compared to non–treated RIL^pur–W22^ seedlings, while the expression levels in RIL^bro–W22^ either remained unchanged or changed to a lesser extent (Appendix A). Similarly, specific DEGs were identified in the NT–RIL^bro–W22^ vs. salt–RIL^bro–W22^ group, and consistent findings were observed in RIL^pur–W22^ (Appendix A).

To gain further insights into the functional pathways of the DEGs specifically expressed in RIL^pur–W22^ and RIL^bro–W22^ seedlings, we conducted a GO enrichment analysis on the 3203 DEGs identified in these two groups (Appendix A). The analysis revealed that the 887 specific, upregulated DEGs in the NT–RIL^pur–W22^ vs. salt–RIL^pur–W22^ group were significantly enriched in functional pathways related to single biological metabolic processes (GO: 0044710), redox processes (GO: 0055114), oxidoreductase activity (GO: 0016491), and the plasma membrane (GO: 0005886) (Figure 4a). On the other hand, the 841 specific, downregulated DEGs were primarily enriched in functional pathways associated with the regulation of anabolism, such as biological regulation (GO: 0065007), the regulation of cellular processes (GO: 0050794), and the regulation of primary metabolic processes (GO: 0080090) (Figure 4b). Additionally, under salt stress, RIL^pur–W22^ seedlings exhibited the induction of genes involved in the anthocyanin synthesis pathway, leading to anthocyanin accumulation (Figure 4b). The upregulation of genes related to redox processes and oxidoreductase activity in RIL^pur–W22^ seedlings after salt treatment further indicates the enhancement of salt tolerance, as it promotes the accumulation of antioxidant substances.

Furthermore, we observed that the GO enrichment analysis of the 818 specific, upregulated DEGs in the NT–RIL^bro–W22^ vs. salt–RIL^bro–W22^ group revealed significant enrichment in functional pathways related to cell metabolism (GO: 0044237), nitrogen compound metabolism (GO: 0006807), chloroplast (GO: 0009507), photosynthesis (GO: 0009768), and quercetin 7–*O*–glucosyltransferase activity (GO: 0080044) (Figure 4c, Appendix A). On the other hand, the 657 specific, downregulated DEGs showed significant enrichment in functional pathways associated with biological regulation (GO: 0065007), the nucleus (GO: 0005634), gene expression regulation (GO: 0010468), ion transport (GO: 0006811), and the regulation of developmental processes (GO: 0050793) (Figure 4d, Appendix A). Notably, the enrichment analysis of upregulated DEGs in RIL^bro–W22^ seedlings revealed the enrichment of genes related to quercetin 3–*O*–glucosyltransferase activity and quercetin 7–*O*–glucosyltransferase activity. These enzymes are involved in the glycosylation of unstable anthocyanins in the flavonoid synthesis pathway, resulting in the production of stable and antioxidant quercetin [40,41]. Additionally, the enrichment of downregulated DEGs in genes related to the regulation of developmental processes indicates that the growth and development of RIL^bro–W22^ seedlings were hindered under salt stress. Moreover, the results of a qRT–PCR analysis of seven DEGs that regulate the developmental process and the seedling growth phenotype of RIL^bro–W22^ under salt treated also confirmed this point (Figure 3d).

In addition, to investigate additional functional pathways that may contribute to the salt tolerance of RIL^bro–W22^ under salt–treated conditions, we conducted a GO enrichment analysis on 140 DEGs (65 upregulated DEGs and 75 downregulated DEGs) specifically responsive to salt stress in the salt–RIL^pur–W22^ vs. salt–RIL^bro–W22^ group (Figure 2e, Appendix A). The enrichment analysis revealed that the upregulated DEGs, in addition to being enriched in functional pathways related to single–cell biological processes, oxidoreductase activity, and redox processes, were also significantly enriched in functional pathways associated with ester bond hydrolase activity (GO: 0016788), glycosyltransferase activity (GO: 0016757), and the carbohydrate metabolism process (GO: 0005975) (Figure 5a, Appendix A). On the other hand, the downregulated DEGs showed significant enrichment in functional pathways related to the plasma membrane (GO: 0005886), iron ion binding (GO: 0005506), signal transduction (GO: 0007165), cell response to stimulation (GO: 0051716), and transport activity (GO: 0005215) (Figure 5b, Appendix A). The enrichment of these functional categories in the upregulated genes suggests potential mechanisms underlying the enhanced salt tolerance of RIL^bro–W22^. Additionally, it was observed that the RIL^bro–W22^ seedlings exhibited greater damage under stress conditions compared to RIL^pur–W22^ seedlings, particularly affecting the plasma membrane system.

### 2.6. Analysis of DEGs Related to Anthocyanin Biosynthesis in RILs

To further investigate the role of anthocyanin accumulation in RIL^pur–W22^ and RIL^bro–W22^, we focused on 44 DEGs associated with the anthocyanin biosynthesis metabolic pathway. Analysis of these DEGs revealed that they could be classified into 11 types of enzymes involved in the anthocyanin synthesis pathway. These enzymes catalyze the conversion of phenylalanine into stable anthocyanins. To visualize the expression patterns of these genes, we normalized the gene expression data across all samples and generated a heat map. The results demonstrated that the majority of genes encoding enzymes in the anthocyanin synthesis pathway were strongly induced in salt–treated RIL^pur–W22^ seedlings, exhibiting higher expression levels than in non–treated RIL^pur–W22^ seedlings. Interestingly, some genes were also slightly induced in salt–treated RIL^bro–W22^ seedlings. Notably, the genes *Zm00001d034635* and *Zm00001d001960*, which encode CHI enzymes, showed even stronger induction in RIL^bro–W22^ seedlings compared to RIL^pur–W22^ seedlings (Figure 6a). In the previous qRT–PCR verification results, we found that the anthocyanin–synthesis–pathway–related genes *Zm00001d014914* (*a2*), *Zm00001d018181* (*fls2*), and *Zm00001d047425* (*F3′5′H*) were upregulated in RIL^pur–W22^ under salt stress (Figure 3c). At the same time, the results of previous studies in our laboratory have shown that salt stress can induce the expression of the anthocyanin–synthesis–pathway–related genes *F3H*, *DFR,* and *ANS* [20], and the expression levels of these three genes in RIL^pur–W22^ seedlings are always higher than those in RIL^bro–W22^, regardless of whether they are untreated or salt–treated (Figure 6).

Furthermore, anthocyanin serves as a non–enzymatic scavenger of reactive oxygen species (ROS) and can effectively neutralize excessive ROS under stress conditions, thereby contributing to salt tolerance in seedlings [42,43]. In the case of RIL^pur–W22^ seedlings exposed to salt stress, the genes involved in the anthocyanin synthesis pathway were induced, resulting in increased anthocyanin accumulation in the seedlings (Figure 6). Therefore, it can be speculated that RIL^pur–W22^ seedlings enhance their tolerance to salt stress by accumulating anthocyanins and participating in the scavenging of reactive oxygen species.

## 3. Discussions

### 3.1. The Salt Stress Tolerance in Maize Seedlings May Be Mediated by Plant Hormones

Hormones play a crucial role in regulating plant responses to abiotic stress. Numerous studies have demonstrated the involvement of hormones in mediating plant tolerance to stress conditions [44,45,46,47]. In our study, transcriptome analysis revealed that a significant number of DEGs was enriched in the “plant hormone signaling” pathway when comparing non–treated and salt–treated RIL^pur–W22^ and RIL^bro–W22^ seedlings. Additionally, through KEGG pathway enrichment analysis, we observed the enrichment of precursor synthesis or metabolic pathways related to various plant hormones [48,49,50,51]. For instance, the “tryptophan biosynthesis” pathway is associated with auxin, the “methionine metabolism” pathway corresponds to ethylene, the “carotenoid synthesis” pathway is related to abscisic acid (ABA), and the “linoleic acid metabolism” pathway is correlated with jasmonic acid (Jas) (Appendix A). These findings indicate the involvement of multiple hormone pathways in the response of RIL^pur–W22^ and RIL^bro–W22^ seedlings to salt stress.

Under abiotic stress conditions, plants exhibit the upregulation of genes related to ABA biosynthesis pathway. High levels of ABA can induce the accumulation of H_2_O_2_ in chloroplasts, leading to stomatal closure and enhancing plant resistance to abiotic stress [52]. Additionally, gibberellins (GAs) can regulate seed germination and seedling growth tolerance to abiotic stress by maintaining ROS homeostasis, thereby increasing the germination rate and productivity [53]. IAA plays a crucial role in plant tolerance to abiotic stresses. For instance, loss of function of the *OsIAA20* gene in rice reduces salt tolerance, affecting yield and seed viability [54]. In apples, the degradation of the MdIAA26 protein by auxin promotes anthocyanin accumulation in fruits [55]. In this study, DEGs enriched in the “plant hormone signal transduction” pathway were analyzed. It was found that of the 98 DEGs, 38 encoded proteins related to indole–3–acetic acid (IAA), 13 encoded ABA–related proteins, 12 encoded ethylene–related proteins, and 16 encoded proteins related to Jas, salicylic acid (SA), brassinosteroids (BR), cytokinins (CTK), and GA (Figure 7a). The cluster heat map of the above hormone–related DEGs was drawn, and it was found that only 15 IAA–related genes were upregulated, while the remaining 23 were downregulated (Figure 7b). Similarly, there were six upregulated and seven downregulated DEGs related to ABA, five upregulated and seven downregulated DEGs related to ACC, one upregulated and five downregulated DEGs related to Jas, and three upregulated and two downregulated DEGs related to SA (Figure 7c–f). Based on the results of previous studies and the transcriptome analysis performed in this study, we speculated that the biosynthesis and signal transduction of plant hormones are inhibited during the development of RIL^bro–W22^ seedlings under salt stress conditions. As a result, RIL^bro–W22^ seedlings may experience difficulty in efficiently removing excessive ROS accumulation, leading to reduced salt tolerance.

### 3.2. Mining Transcription Factors That Potentially Regulate Salt Stress Tolerance in Maize Seedlings

Previous studies have demonstrated the involvement of transcription factors belonging to the bZIP, NAC, MYB, and WRKY families in regulating plant responses to salt stress [56,57,58,59]. In this study, we retrieved a total of 2194 transcription factors from the Plant Transcription Factor Database (http://planttfdb.gao–lab.org/index.php, accessed on 26 June 2023). And we identified 472 and 420 DEGs encoding transcription factors in the NT–RIL^pur–W22^ vs. salt–RIL^pur–W22^ and NT–RIL^bro–W22^ vs. salt–RIL^bro–W22^ groups, respectively (Figure 8a). KEGG analysis revealed the significant enrichment of DEGs encoding transcription factors in the “plant hormone signal transduction” and “plant MAPK signaling pathway” categories for both groups (Appendix A). These results further support the involvement of plant hormones in mediating salt stress tolerance in maize seedlings. Notably, studies by Cai et al. have demonstrated that maize ZmWRKY17 negatively regulates ABA signal transduction to enhance salt tolerance in seedlings [60], while rice OsWRKY50 gene positively regulates ABA–independent signaling to enhance salt tolerance [61]. These findings further corroborate our speculations.

Further analysis focused on the 46 enriched transcription factors (Figure 8b). Among them, twelve DEGs belonged to the bZIP family, ten belonged to the WRKY family, and the remaining belonged to ERF (five DEGs), bHLH (three DEGs), and TCP (three DEGs) families. By comparing the expression patterns of these 46 transcription factors between non–treated and salt–stressed RIL^pur–W22^ and RIL^bro–W22^ seedlings, we observed significant changes in expression levels in RIL^pur–W22^ seedlings under salt stress compared to non–treated RIL^pur–W22^ seedlings. Similarly, DEGs exclusively expressed in the NT–RIL^bro–W22^ vs. salt–RIL^bro–W22^ group showed similar expression patterns in RIL^bro–W22^ seedlings under different treatments (Figure 8c–e). These findings suggest that different transcription factors may positively regulate seedling tolerance to salt stress while others may exert negative regulatory roles. This also provides new insights and candidate genes for further research on transcription factors to enhance the salt tolerance of maize seedlings by regulating anthocyanin–synthesis–related genes in the future.

## 4. Materials and Methods

### 4.1. Evaluation of Plant Materials and Traits

The purple W22 (pur–W22) inbred line was obtained from the laboratory of Professor Mingliang Xu, International Maize Improvement Center, China Agricultural University. The bronze W22 (bro–W22) inbred line was obtained from the mutant maize COOP stock center. The homozygous RIL^pur–W22^ and RIL^bro–W22^ materials were selected from the cross–pollination (or reciprocal cross) of pur–W22/bro–W22 for generations in both Sanya (Hainan province) and Zhuozhou (Hei Bei province), in winter and summer, respectively.

Standard germination: 90 seeds (3 replicates, 30 in each replicate) were disinfected in 0.1% sodium hypochlorite for 5 min, rinsed with distilled water three times, and then sown in germination paper. The rolled germination paper was placed vertically in a sealed plastic bag and cultured in an incubator with a light/dark cycle of 16 h/8 h at 25 °C.

Salt–treated paper germination: 90 seeds were disinfected using standard germination steps, seeded on germination paper containing 100 mM NaCl solution, and then cultured according to standard germination culture conditions.

Cultivation in sand: 90 seeds were sterilized in 0.1% sodium hypochlorite solution, washed three times with distilled water, and sowed in sandy soil containing 0 mM and 100 mM NaCl solutions. The germination box after sowing was placed in an incubator at 25 °C and cultured at a 16 h/8 h light–dark ratio for 14 days. Stem length and root length were measured with a ruler 14 days after sowing. The histogram of root length and seedling length was drawn using the Graphpad Prism 8 software package, and the *p* value was calculated by one–way analysis of variance.

### 4.2. Phenotypic Analysis of Seed Morphology

Fifty randomly selected seeds were scanned three times repeatedly using an EPSON J221A scanner (Seiko Epson Corporation, Suga, Nagano–ken, Japan), and then the data regarding seed length, seed width, and seed thickness were analyzed using the Seeds Identification and Photoshop software packages [62]. Finally, the Graphpad Prism 8 software package was used to plot the measured data, and one–way analysis of variance was used to calculate the *p* value.

### 4.3. RNA Extraction and Sequencing

RIL^pur–W22^ and RIL^bro–W22^ seedlings (including shoots and roots, with the seeds removed) germinated for 14 days in 0 mM and 100 mM NaCl solution environments for RNA sequencing, with two replicates for each sample. Total RNA was extracted using the RNA extraction kit (Mei5bio, Beijing, China). RNA concentration and quality were measured using an ultra–micro spectrophotometer ND2000 (Thermo Scientific, New York, NY, USA). The prepared RNA was sent to Annoroad Gene Technology (Beijing, China) for library construction, sequencing, and data filtering (Appendix A). Using the DNBSEQ–T7 sequencing platform, and the libraries were sequenced with a read length of 150 bp (pair–end).

### 4.4. Sequence Data Analysis

The filtered data were used to establish a genome index file using the Hisat2 program and were compared to the maize reference genome (B73_RefGen_v4) [63]. The Samtools program was used to convert the compared.sam format file into a .bam format file for subsequent data call and analysis. Transcripts were assembled and gene expression was estimated using the Stringtie and FeatureCounts programs. Differentially expressed genes were analyzed using the R language program DESeq2 (https://bioconductor.org/packages/release/bioc/html/DESeq2.html, accessed on 26 June 2023) [64]. When analyzing the RNA–Seq data of non–treated and salt–treated seedlings, the screening conditions were an adjusted *p*–value of <0.01 and an absolute difference multiple fold change (FC) of ≥2. The FPKM value is calculated using the number of raw reads, and the formula is: FPKM = read count/(mapping reads (millions)) × exon length (KB). If FPKM ≥ 1 in at least one sample, the gene was considered to be an expressed gene. All of the expressed genes from different samples were applied to TBtools for PCA calculation, using the default settings [65].

### 4.5. Cluster Analysis and Functional Annotation Enrichment Analysis

The obtained differentially expressed genes were subjected to gene ontology (GO) enrichment analysis using agriGO v2.0 (agriGO single species analysis) [66]. After the gene ID was converted using MaizeGDB (https://www.maizegdb.org/gene_center/gene, accessed on 26 June 2023), KOBAS (http://kobas.cbi.pku.edu.cn/, accessed on 26 June 2023) was used for Kyoto Encyclopedia of Genes and Genomes (KEGG) pathway enrichment analysis [67].

### 4.6. qRT–PCR Analysis

First–strand cDNA was synthesized using a StarScript II RT Mix with gDNA Remover kit (GenStar, Beijing, China). qRT–PCR was performed using 2 × HQ SYBR qPCR Mix (Zomanbio, Beijing, China) in triplicate on an ABI Life Q6 real–time fluorescent quantitative PCR instrument (Applied Biosystems, Waltham, MA, USA) and using the obtained results and the 2^–Δ^CT method for relative quantification. The *GAPDH* was used as an internal control. The primer sequences used are listed in Appendix A.

## 5. Conclusions

We screened two maize W22 inbred lines (salt–tolerant line pur–W22 and sensitive line bro–W22) regarding the different salt tolerance levels of seedlings under a salt stress environment and performed transcriptome analysis of their RILs to identify the genes related to the salt stress tolerance of maize seedlings. We found that the specific expression of DEGs in the salt tolerance of RIL^pur–W22^ seedlings was mainly related to the redox process, biological regulation, and the plasma membrane. Among them, anthocyanidin–synthesis–related genes were strongly induced under salt treatment, which was partially consistent with the physiological results regarding salt tolerance in seedlings. The results showed that improving the anthocyanidin–synthesis–related genes in maize could effectively compensate for seedling growth inhibition caused by salt stress. This study could lay a foundation for mining and cloning key genes affecting the salt tolerance of maize seedlings under salt stress conditions.

## Figures and Tables

**Figure 1 plants-12-02793-f001:**
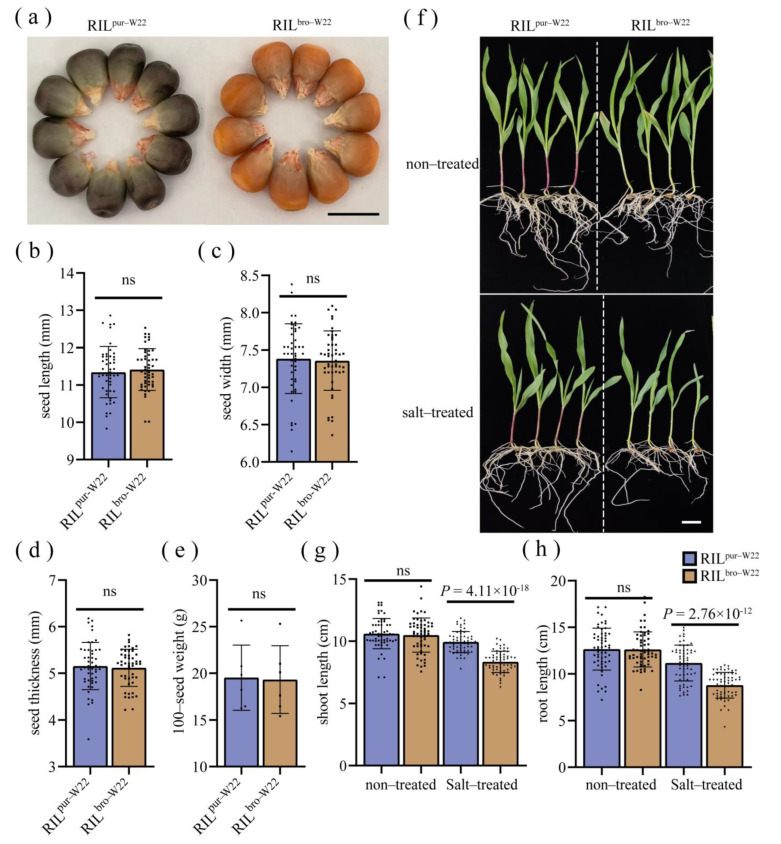
Seeds and seedling characteristics of RIL^pur–w22^ and RIL^bro–w22^. (**a**–**e**) Seed phenotypes of two RILs and their (**a**), seed lengths (**b**), seed widths (**c**), seed thicknesses (**d**), and 100–seed weights (**e**). (**f**–**h**) Seedling phenotypes (**f**), shoot lengths (**g**), and root lengths (**h**) of two RILs grown for 14 days under non–treated and salt–treated (100 mM NaCl solution) conditions. Bar = 1 cm; black dots, squares and triangles represent individual values for different samples; *p* values calculated by one–way ANOVA; *p* < 0.01 indicates that the difference is extremely significant; ns represents no difference.

**Figure 2 plants-12-02793-f002:**
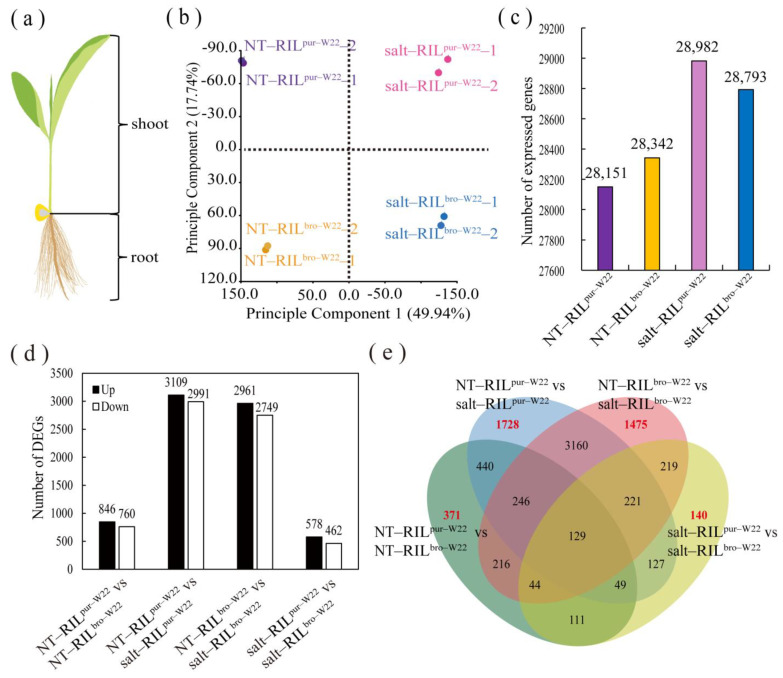
Seedling transcriptome analysis of RIL^pur–W22^ and RIL^bro–W22^ under non–treated and salt–treated conditions. (**a**) Seedling sample schematic for RNA extraction. (**b**) Principal component analysis of the gene expression profiles of RIL^pur–W22^ and RIL^bro–W22^ seedlings under non–treated and salt–treated conditions. (**c**) The number of expressed genes (FPKM ≥ 1) identified in non–treated RIL^pur–W22^ (NT–RIL^pur–W22^), non–treated RIL^bro–W22^ (NT–RIL^bro–W22^), salt–treated RIL^pur–W22^ (salt–RIL^pur–W22^), and salt–treated RIL^bro–W22^ (salt–RIL^bro–W22^). (**d**) Comparison of the number of upregulated and downregulated DEGs in the NT–RIL^pur–W22^ vs. NT–RIL^bro–W22^, NT–RIL^pur–W22^ vs. salt–RIL^pur–W22^, NT–RIL^bro–W22^ vs. salt–RIL^bro–W22^, and salt–RIL^pur–W22^ vs. salt–RIL^bro–W22^ groups. (**e**) Venn diagram drawn using the DEGs in the NT–RIL^pur–W22^ vs. NT–RIL^bro–W22^, NT–RIL^pur–W22^ vs. salt–RIL^pur–W22^, NT–RIL^bro–W22^ vs. salt–RIL^bro–W22^ and salt–RIL^pur–W22^ vs. salt–RIL^bro–W22^ groups. Red numbers refer to DEGs specifically expressed in the above groups.

**Figure 3 plants-12-02793-f003:**
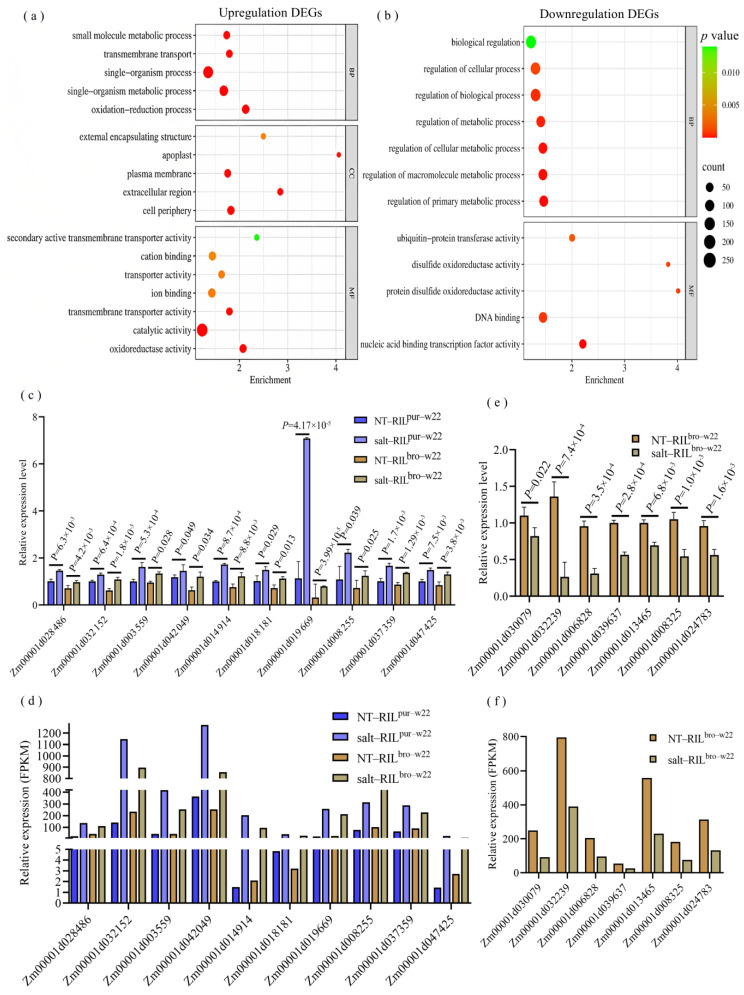
GO enrichment analysis of 3160 DEGs’ common response to salt stress in RIL^pur−W22^ (NT−RIL^pur−W22^ vs. salt−RIL^pur−W22^) and RIL^bro–W22^ (NT–RIL^bro–W22^ vs. salt–RIL^bro–W22^) seedlings, of which 1637 DEGs were upregulated (**a**) and 1521 DEGs were downregulated (**b**). The size and color scale of the points in the figure represent the number and significance level of DEGs in GO terms, respectively. (**c**,**d**) qRT–PCR (**c**) and RNA–seq (**d**) analysis of the DEGs with regard to oxidoreductase activity. (**e**,**f**) qRT–PCR (**e**) and RNA–seq (**f**) analysis of the DEGs with regard to the regulation of the developmental process. *GAPDH* was used as an internal control. Values are shown as means ± SE, *n* = 3; *p* values calculated by one–way ANOVA; *p* < 0.05 represents a significant difference; *p* < 0.01 indicates that the difference is extremely significant.

**Figure 4 plants-12-02793-f004:**
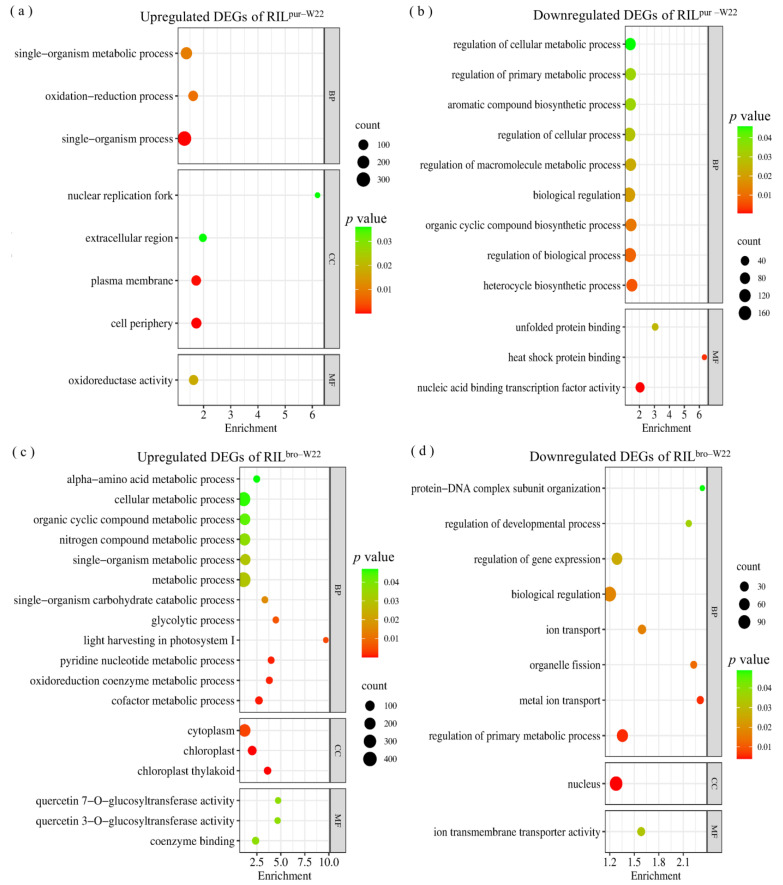
GO analysis of the 1728 DEGs which specifically respond to salt stress in RIL^pur–W22^ (NT–RIL^pur–W22^ vs. salt–RIL^pur–W22^) seedlings. The left side refers to the 887 upregulated DEGs (**a**), and the right side refers to the 841 downregulated DEGs (**b**). GO analysis of the 1475 DEGs which specifically respond to salt stress in RIL^bro–W22^ (NT–RIL^bro–W22^ vs. salt–RIL^bro–W22^) seedlings. The left side refers to the 818 upregulated DEGs (**c**), and the right side refers to the 657 downregulated DEGs (**d**). The size and color scale of the points in the figure represent the number and significance level of DEGs in GO terms, respectively.

**Figure 5 plants-12-02793-f005:**
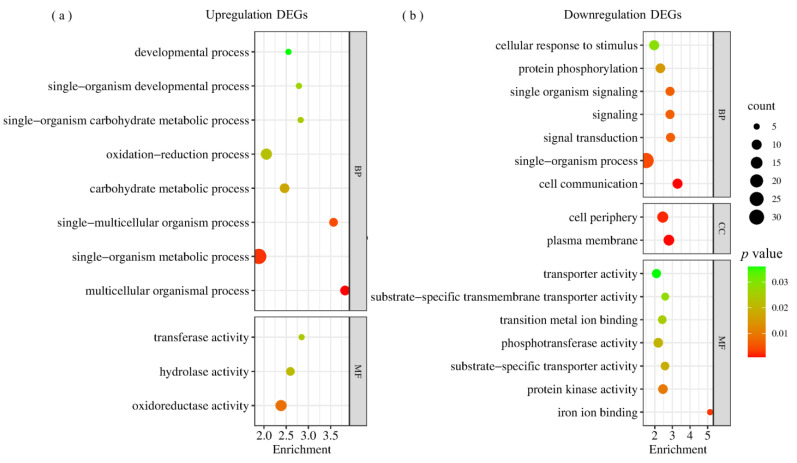
GO analysis of the 140 DEGs which specifically respond to salt stress in salt–treated RIL^pur–W22^ and RIL^bro–W22^ (salt–RIL^pur–W22^ vs. salt–RIL^bro–W22^) seedlings. The left side refers to the 65 upregulated DEGs (**a**), and the right side refers to the 75 downregulated DEGs (**b**). The size and color scale of the points in the figure represent the number and significance level of DEGs in GO terms, respectively.

**Figure 6 plants-12-02793-f006:**
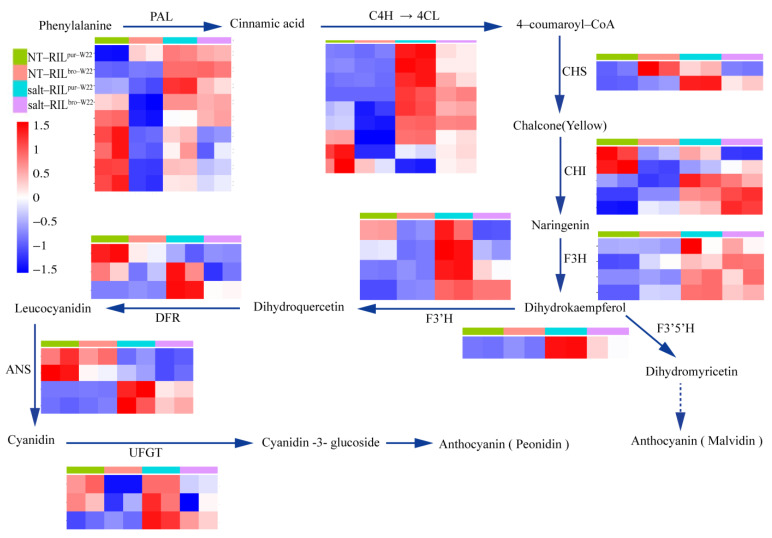
Clustering heatmap of genes related to anthocyanin synthesis pathway in expressed genes. Each sample has two compartments, which are two biological replicates. According to the standardized FPKM, red and blue indicate high and low abundance, respectively.

**Figure 7 plants-12-02793-f007:**
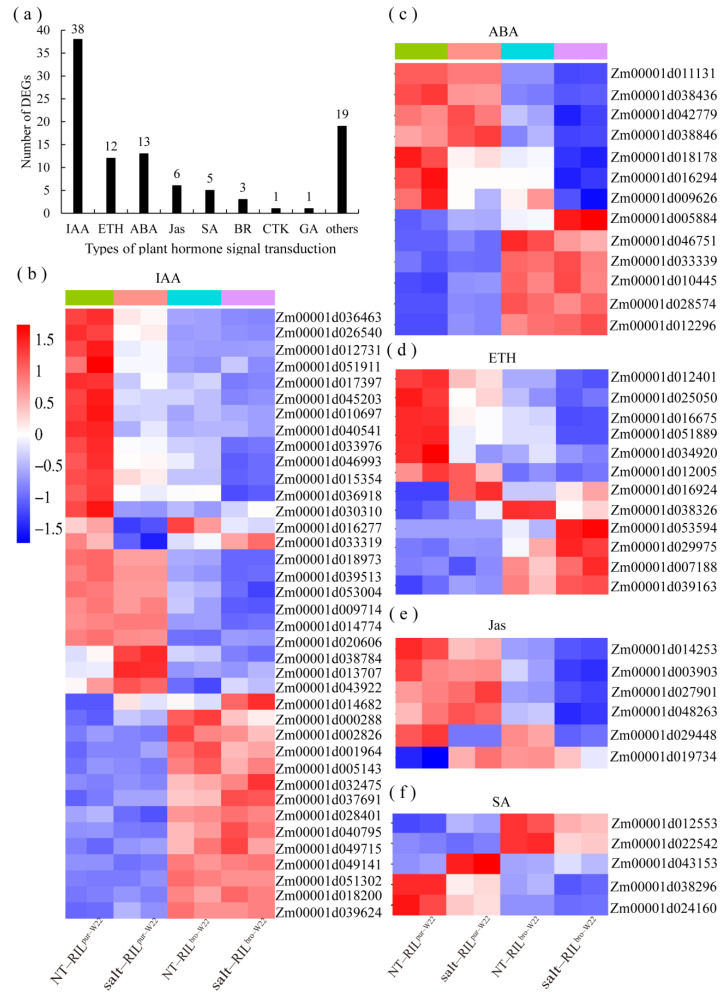
DEGs identified in “plant hormone pathways”. (**a**) The types of hormones in the “plant hormone pathway” and the number of DEGs belonging to these types. (**b**–**f**) Heatmap clustering of the 46 DEGs related to auxin (IAA, (**b**)), abscisic acid (ABA, (**c**)), ethylene (ETH, (**d**)), Jas monic acid (Jas, (**e**)), and salicylic acid (SA, (**f**)) in non–treated and salt–treated RIL^pur–W22^ and RIL^bro–W22^ seedlings (NT–RIL^pur–W22^, salt–RIL^pur–W22^, NT–RIL^bro–W22^, and salt–RIL^bro–W22^). Each sample has two compartments, which are two biological replicates. According to the standardized FPKM, red and blue indicate high and low abundance, respectively.

**Figure 8 plants-12-02793-f008:**
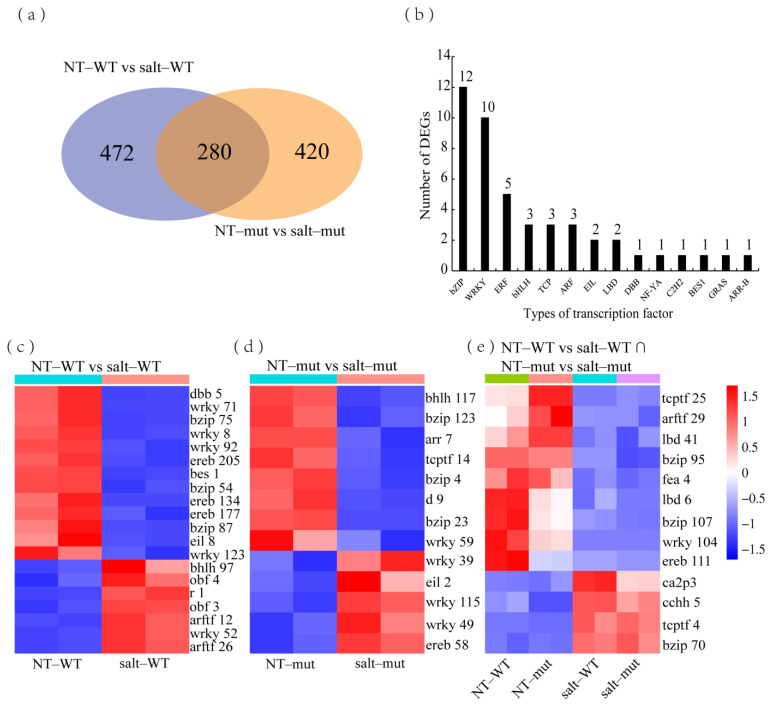
Analysis of DEGs encoding transcription factors. (**a**) DEGs encoding transcription factors in the NT–RIL^pur–W22^ vs. salt–RIL^pur–W22^ and NT–RIL^bro–W22^ vs. salt–RIL^bro–W22^ groups. (**b**) Transcription factor types and numbers. (**c**) Heatmap of 20 transcription factors which exhibit special expression in non–treated and salt–treated RIL^pur–W22^ (NT–RIL^pur–W22^ vs. salt–RIL^pur–W22^) seedlings (seven upregulated and thirteen downregulated). (**d**) The heatmap of thirteen transcription factors which special expression in non–treated and salt–treated RIL^bro–W22^ (NT–RIL^bro–W22^ vs. salt–RIL^bro–W22^) seedlings (five upregulated and right downregulated). (**e**) The heatmap of thirteen transcription factors which common expression in non–treated and salt–treated RIL^pur–W22^ and RIL^bro–W22^ (NT–RIL^bro–W22^ vs. salt–RIL^bro–W22^ ∩ NT–RIL^bro–W22^ vs. salt–RIL^bro–W22^) seedlings (four upregulated and nine downregulated). Each sample has two compartments, which are two biological replicates. According to the standardized FPKM, red and blue indicate high and low abundance, respectively.

**Table 1 plants-12-02793-t001:** Transcriptome sequencing data statistics.

Lines	Rep	Total Reads	Rate of Total Mapped Reads (%)	Num. of Expressed Genes	Rate of Expressed Genes (%)
NT–RIL^pur–W22^	1	47,540,654	96.8	28,187	60.72
	2	47,933,524	97.04	28,115	60.56
NT–RIL^bro–W22^	1	44,597,506	97.33	28,317	61.00
	2	45,014,472	96.00	28,367	61.10
salt–RIL^pur–W22^	1	47,815,598	96.41	28,991	62.45
	2	46,306,714	95.15	28,974	62.41
salt–RIL^bro–W22^	1	45,334,562	96.63	28,886	62.22
	2	45,274,946	95.58	28,701	61.82

Note: Total reads: the number of reads after filtering the original data; rate of total mapped reads (%): the ratio of the remaining reads after filtering to the original unfiltered reads; num. of expressed genes: total number of expressed genes with FPKM ≥ 1; rate of expressed genes (%): the proportion of the total number of expressed genes with FPKM ≥ 1 to the total number of genes.

## Data Availability

The data that support the findings of this study are available at www.ncbi.nlm.nih.gov/geo with accession number PRJNA986440.

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
