# Peer review of "Transcriptome Analysis Revealed the Potential Molecular Mechanism of Anthocyanidins’ Improved Salt Tolerance in Maize Seedlings"

_plants, 2023, doi:10.3390/plants12152793_

Round 1

Reviewer 1 Report

This work by Wang et. investigated the transcriptome profile of two maize RILs with different seed colors. Although the study provides some interesting insights into the regulation of anthocyanin or flavonoid in maize seedlings, there are several concerns that need to be addressed by the authors.

I have provided my comments in the PDF manuscript file attached below.

There are no major English or grammatical issues. However, the authors should closely check for semantic throughout the manuscript.

Author Response

1” MYB, bHLH, WD40” spell in full when mentioned for the first time in the manuscript

In the manuscript, MYB, bHLH, WD40 were changed to v-myb avian myeloblastosis viral oncogene homolog (MYB), basic helix-loop-helix (bHLH), WD-repeat protein (WD40)

2 Furthermore, environmental factors also impact the accumulation of anthocyanins in plants [20-24]. Under environmental stress, plants can regulate the expression of structural genes and regulatory genes in the anthocyanin synthesis pathway to accumulate anthocyanins in plants to withstand the damage caused by environmental stress [25-28]. This paragraph requires improvement to provide a good flow and clear understanding. The authors should also focus on changes in anthocyanin under salt stress

This section has been modified and an example of anthocyanin changes under salt stress has been added. The content of the supplement was marked with yellow backgrounded.

3 which exhibit different level of salt tolerance in seedlings. The findings not only enhance our understanding of the factors contributing to the contrasting salt tolerance of seedlings between the two W22 inbred lines but also identify new candidate genes for breeding maize varieties with high anthocyanin content and strong salt tolerance. The authors failed to portray a clear hypothesis on the essence of this study and what is expected and eventually how to achieve that. This is not a place to discuss the findings and their significance. Must be revised.

This section has been modified in the manuscript, and the modified content was marked with yellow backgrounded.

4 The seeds of the purple W22 (pur-W22) inbred lines exhibited a purple color, while the seeds of the bronze W22 (bro-W22) inbred lines had a bronze color.  This statement does not provide insight into the essence of these results. Is this observation a result of this experiment? which I think it is not. Improve the description of the results here by maintaing a certain logic for better understading of the results

The logic of this section was to observe the seed phenotype of the two inbred lines, and found that the parents had differences in seed length, seed width, seed thickness and 100-seed weight in addition to seed color. The salt tolerance of parent seedlings was evaluated, and it was found that pur-W22 had stronger salt tolerance. Therefore, it is impossible to determine the factors that cause its salt tolerance. Therefore, by constructing RILs population, the differences caused by seed length, seed width, seed thickness and 100-seed weight were removed, and then whether the two inbred lines had different salt stress tolerance was evaluated. According to this logic, the result description is modified and supplemented, and the modified places were marked with yellow backgrounded.

5 To evaluate the salt stress tolerance of the pur-W22 and bro-W22, we subjected them to an 100 mM NaCl solution to simulate salt stress. Both lines exhibited similar germination rates, exceeding 96%, under both non-treated and salt-treated conditions (Figure S1g). This statment should not be here. Describe the results with regard to the treatment directly

“Both lines exhibited similar germination rates, exceeding 88 96%, under both non-treated and salt-treated conditions (Figure S1g) ”. The position of this sentence has been adjusted, and the specific adjustments have been highlighted with yellow backgrounded in the manuscript

6 Figure 1. Use the same annotation for the p-value,Most of panels show non significant results between the two lines. There is no need to show these results here. Remove and keep those with clear statistical significant difference; and indicate that in the manuscript.

Panel "f" should also describe changes in the phenotype of each line untreated and treated  to see the effect within the line.

The value of P > 0.05 in Figure 1 was deleted, and ns was used to indicate that there was no difference between the two samples. The results of the comparison of non-treated and salt-treated in RILpur-w22 and RILbro-w22 seedlings in Figure 1f has been supplemented in the manuscript, and the supplementary content is marked in red font.

7 .95.15%-97.33% change dash sign to minus sign to indicate the range

We modify “95.15% -97.33%” to “95.15% to 97.33%”

8 Genes with an FPKM value ≥ 1 w Several studies consider DEGs with fold change equal or above 2. Why did the authors considered 1 fold changee?

FPKM (Fragments Per Kilobase Million) is a commonly used method for standardizing gene expression levels, which takes into account the effects of sequencing depth and gene length on gene expression count. Usually, if FPKM ≥ 1 in at least one sample, the gene is considered to be an expressed gene. Here we indicated an expression gene, but not a fold changed gene.

9 Using a threshold change of ≥ 2 and a corrected p-value of ≤ 0.01, we screened  differentially expressed genes (DEGs) in various comparisons. The comparisons included non-treated RILpur-W22 vs RILbro-W22 (NT-RILpur-W22 vs NT-RILbro-W22), non-treated RILpur-W22 vs salt-treated RILpur-W22 (NT-RILpur-W22 vs salt-RILpur-W22), non-treated RILbro-W22 vs salt-treated RILbro-W22 (NT-RILbro-W22 vs salt-RILbro-W22), and salt-treated RILpur-W22 vs RILbro-W22 (salt-RILpur-W22 vs salt-RILbro-W22) (Figure 2d). The details are not necessary here. Go straight to the results and describe them

“The comparisons included non-treated RILpur-W22 vs RILbro-W22 (NT-RILpur-W22 vs NT-RILbro-W22), non-treated RILpur-W22 vs salt-treated RILpur-W22 (NT-RILpur-W22 vs salt-RILpur-W22), non-treated RILbro-W22 vs salt-treated RILbro-W22 (NT-RILbro-W22 vs salt-RILbro-W22), and salt-treated RILpur-W22 vs RILbro-W22 (salt-RILpur-W22 vs salt-RILbro-W22) (Figure 2d).” has been removed from the manuscript.

10 Figure 3. GO enrichment analysis of the common up-regulated DEGs  Validation results usually refers to RNA-seq results versus qPCR. Here, the authors only showed qPCR without RNA-seq results of selected genes. Need to show. Also, add a table of top most 10 up- and downregulated DEGs in both backgrounds

The RNA-seq results of qRT-PCR genes were supplemented in Figure 3 (Figure 3 d, f). In addition, the up-regulated and down-regulated DEGs in RILpur-W22 and RILbro-W22 in response to salt stress were listed in Supplementary Table S1.

11 2.5 Specific response of RILpur-W22 and RILbro-W22 to salt stress this result title is not informative. consider providing a meaningful title

We Changed ” Specific response of RILpur-W22 and RILbro-W22 to salt stress” to “Specific salt-induced DEGs in RILpur-W22 and RILbro-W22

12 Considering the complexity of the study and the results, the authors should add a conclusion that portray their findings and give a take-home message for the readers

We have updated the conclusion already.

Reviewer 2 Report

Dear Editor,

Thank you for choosing me as a reviewer of the manuscript plants-2497452 entitled: ”Transcriptome Analysis Revealed the Anthocyanidins Improved Salt Tolerance in Maize Seedlings” I hope that my comments will help authors to improve their manuscript.

Detailing remarks concerning the manuscript.

1.       It is not recommended to use as key words the words or phrases used in the title of the manuscript. Please do needed changes.

2.       In the ‘Introduction’ section there is no even short information why the salt stress was chosen in the studies.

3.       All the figures and tables should be clear for the reader without the referring to the text of the manuscript. Please give the appropriate explanations where needed.

4.       The clear purpose of the report together with scientific hypothesis and the answer to the question stated as scientific hypothesis should be given

5.       Please verify the titles of the figures see: ‘Figure 8. Analysis of the DEGs whicFigure 8. Analysis of DEGs encoding transcription factors...’

6.       I suggest to verify the description of the ‘Results’ and ‘Discussion’ section. Some of the results (figures 7 and 8) are presented in the Discussion section.

7.       References should be prepared strictly to the guidelines for authors. There are editorial mistakes that should be improved. For example once each word of the manuscript title is written with capital letter, but the other time only the first word of the manuscript title is written with capital letter. Please go carefully through the whole references improving all editorial mistakes.

8.       The Latin names of the species should be italicized. Please do needed changes.

9.       The conclusions are missed

Author Response

  1. It is not recommended to use as key words the words or phrases used in the title of the manuscript. Please do needed changes.

Thanks, the title and also key words have been changed. The title “Transcriptome Analysis Revealed the Anthocyanidins Im-proved Salt Tolerance in Maize Seedlings” has been changed into “Transcriptome Analysis Revealed the Potential Molecular Mechanism of Anthocyanidins Improved Salt Tolerance in Maize Seedlings”, and the Keywords, from “maize seedling, salt tolerance, anthocyanidin, transcriptome sequencing” into “maize; seedling growth; salt stress tolerance; anthocyanidin; transcriptome sequencing”.

  1. In the ‘Introduction’ section there is no even short information why the salt stress was chosen in the studies.

The introduction has been updated and we emphasized the study of salt stress separately. The detailed please see the third paragraph in this revised manuscript.

  1. All the figures and tables should be clear for the reader without the referring to the text of the manuscript. Please give the appropriate explanations where needed.

The Figure 3 has been replaced with a high-resolution picture, and in addition to Figure 6, the other Figure legends were supplemented and described, and the specific supplement were marked with yellow highlight.

  1. The clear purpose of the report together with scientific hypothesis and the answer to the question stated as scientific hypothesis should be given

Corrected, please see updated introduction in manuscript.

  1. Please verify the titles of the figures see: ‘Figure 8. Analysis of the DEGs whicFigure 8. Analysis of DEGs encoding transcription factors...’

Corrected, we deleted the redundant words, please see updated introduction in manuscript.

  1. I suggest to verify the description of the ‘Results’ and ‘Discussion’ section. Some of the results (figures 7 and 8) are presented in the Discussion section.

Thanks for the suggestion. Some of the results need to be further testified, so we put them in discussion, to leave foreshadowing.

  1. References should be prepared strictly to the guidelines for authors. There are editorial mistakes that should be improved. For example once each word of the manuscript title is written with capital letter, but the other time only the first word of the manuscript title is written with capital letter. Please go carefully through the whole references improving all editorial mistakes.

All the citations in reference part have been corrected, and marked with yellow highlight.

  1. The Latin names of the species should be italicized. Please do needed changes.

The italic problem was corrected, and marked with yellow highlight.

  1. The conclusions are missed

We have updated the conclusion already.

Reviewer 3 Report

The authors in this study investigated the role of anthocyanin accumulation in enhancing salt tolerance in maize seedlings. The study utilized two W22 inbred lines, RILpur-W22 and RILbro-W22, which exhibit different levels of salt tolerance. Through transcriptome analysis, the study identified differentially expressed genes (DEGs) associated with salt stress response. A total of 6,100 DEGs were identified in RILpur-W22 seedlings, and 5,710 DEGs were identified in RILbro-W22 seedlings, from salt-stressed conditions compared to non-treated groups. Among these DEGs, 3,160 were common to both lines and enriched in various biological processes. Additionally, 1,728 specific DEGs were found in the salt-tolerant line RILpur-W22. The study also investigated the expression patterns of transcription factors and their potential role in seedling salt tolerance. The findings provide valuable insights into the molecular mechanisms of anthocyanin involvement in seedling salt tolerance and offer candidate genes for future research and breeding efforts.

However, I have a couple of questions/suggestions:

·         The manuscript lacks specific information about the protocols and kits used for certain procedures, as well as detailed statistical analysis methods.

·         Why was V4 of the maize genome used while V5 is available?

·         It is unclear which software was used for various analyses. In the materials and methods section, please provide a detailed explanation of the software used for the analysis, along with appropriate citations. Additionally, consider making all the analysis codes publicly available by creating a GitHub repository. This will facilitate better reuse of the analysis and ensure transparency in the analysis process.

There are a few grammatical errors present in the article that can be easily rectified by conducting a thorough review of the manuscript.

Author Response

1. The manuscript lacks specific information about the protocols and kits used for certain procedures, as well as detailed statistical analysis methods.

The steps and kits involved have been checked and nothing is missing. We have added statistical methods to the attached figure and Materials and methods(4.1 and 4.2), and the statistical methods in this paper used one-way analysis of variance.

2. Why was V4 of the maize genome used while V5 is available?

Since our genome was been assembled according to maize genome V4 when we carried out our experiments. We think it acceptable for transcriptome analysis. So far there is no requirements for V5 or V4 in this kind of research.

3. It is unclear which software was used for various analyses. In the materials and methods section, please provide a detailed explanation of the software used for the analysis, along with appropriate citations. Additionally, consider making all the analysis codes publicly available by creating a GitHub repository. This will facilitate better reuse of the analysis and ensure transparency in the analysis process.

We updated the citations in “Material and Methods part” about DESeq2、TBtools、agriGO v2.0 and KOBAS, as see the reference 64-67.

4. Comments on the Quality of English Language: There are a few grammatical errors present in the article that can be easily rectified by conducting a thorough review of the manuscript.

Updated. Thanks very much for the comments.

Round 2

Reviewer 1 Report

The manuscript has been improved by the authors as per my comments.

The manuscript has been improved by the authors as per my comments.